# Why Do Iranian Preschool-Aged Children Spend too Much Time in Front of Screens? A Preliminary Qualitative Study

**DOI:** 10.3390/children10071193

**Published:** 2023-07-10

**Authors:** Bita Shalani, Parviz Azadfallah, Hojjatollah Farahani, Serge Brand

**Affiliations:** 1Department of Psychology, Tarbiat Modares University, Tehran 14117-13116, Iran; bita.shalani@gmail.com (B.S.); h.Farahani@modares.ac.ir (H.F.); 2Center for Affective, Sleep and Stress Disorders, Psychiatric Clinics of the University of Basel, 4002 Basel, Switzerland; 3Division of Sport Science and Psychosocial Health, Department of Sport, Exercise and Health, Faculty of Medicine, University of Basel, 4002 Basel, Switzerland; 4Substance Abuse Prevention Research Center, Kermanshah University of Medical Sciences (KUMS), Kermanshah 67158-47141, Iran; 5Sleep Disorders Research Center, Kermanshah University of Medical Sciences (KUMS), Kermanshah 67158-47141, Iran; 6School of Medicine, Tehran University of Medical Sciences (TUMS), Tehran 14166-34793, Iran; 7Center for Disaster Psychiatry and Disaster Psychology, Psychiatric Clinics of the University of Basel, 4002 Basel, Switzerland

**Keywords:** screen time (ST), qualitative research, context of family system, preschoolers, parents

## Abstract

There is evidence that Iranian preschool children are increasingly spending their time in front of screens (screen time: ST; time spent with any screen such as TVs, computers, tablets, smartphones, game consoles, or video games), but few studies have explored the possible causes of such an increase. Given this, the present study aimed to qualitatively explore determinants of excessive ST in Iranian children. To this end, parents of preschool children were interviewed, and their answers were qualitatively clustered to identify additional important factors. Key informant interviews were conducted with parents of preschool children in Tehran (Iran). A semi-structured interview was developed to assess child and family life, daily routine, family rules, family interactions, and home climate as possible contributing factors to ST. Parents’ audiotaped statements were transcripted verbatim, coded, and clustered into main themes using thematic analysis with the MaxQda^®^ software. A total of 20 parents of children aged 2 to 7 were interviewed, and a total of 6 key themes and 28 subthemes were extracted from their interviews. The results of the analysis identified a broad range of both independent and interrelated factors leading to the development and maintenance of ST behaviors among preschool children. Our findings indicate that the central concept is the family. Considering screen-related behaviors, family life encompasses parental health literacy (e.g., parenting pattern, monitoring standards, thoughtful parenting), family psychological atmosphere (e.g., presence of parents, family norms, parent–parent and parent–child interaction, congruency/incongruency of parents with each other) and the digital structure of the home. The child’s and parents’ actions and characteristics can influence family interactions. A child’s and parent’s behavior is also influenced by social/cultural factors. Parents’ behaviors and attitudes, family communications, and interactions contribute to healthy ST habits in children. It is not possible to examine the child’s behavior without considering the family and the dominant environment, since the behavior of family members as a whole affects each family member. Given this, interventions should make parents aware of their role and responsibilities in reducing children’s ST and consider the family system as a whole, and interventions also can benefit from considering the parental perceptions of children’s behaviors.

## 1. Introduction

The term screen time refers to time spent with any screen or digital media devices, such as TVs, computers, tablets, smartphones, game consoles, or video games [1]. While screens are part of the everyday lives of adolescents and adults, it appears that in particular, children below seven years old show excessive ST [2,3]. However, such high rates of ST among preschoolers appear to be related to a preschooler’s higher scores for developmental, and thus, psychological health concerns [4,5]. Childhood is an important time in a person’s development, as it is a period of rapid brain growth, sensory and motor development, and the development of higher-order functions such as language and affective development. During this time, the brain is particularly sensitive to environmental influences and experiences that can shape later development [6]. In this view, children need natural experiences such as playing outdoors, interacting with peers, and parent–child interaction to experience normative development during critical developmental periods. However, there is evidence that above all, excessive ST may negatively impact a child’s brain development [7]. According to the American Academy of Pediatrics (AAP) and the Canadian Pediatric Society, children aged 2–5 years should not be exposed to ST beyond an hour, and children younger than 2 years should not be exposed to ST at all (except for video chatting) [8,9]. The Australian Department of Health recommends that children and teenagers should not spend more than 30 min in sedentary activities before and after school, and no more than two hours total in vacation time [10]. A recent study has revealed that only 13.9% of children follow all three of the 24 h guidelines recommended by the World Health Organization. This is in stark contrast to the overall global increase in the number of people following screen time (ST) guidelines [11].

There are a number of ways in which ST may unfavorably impact on children’s physical and psychological health: Higher scores for ST were related to higher scores for sleep problems, lack of sleep, anxiety, emotional distress, attention problems, depression, myopia, reduced or light physical activity, and behavioral problems [12,13,14,15,16,17], it also interferences with caregiver–child relationships, communication, learning, development, and wellbeing [17,18,19,20,21]. Similarly, higher scores for physical inactivity due to excessive screen exposure were associated with higher scores for obesity, heart risk factors, lower fitness [22], metabolic diseases, and cardiovascular diseases [23]. Unsurprisingly, Motamed-Gorji, et al. [24] showed that a longer ST duration was associated with lower scores for the total quality of life among Iranian students aged 6 to 18. Moreover, the results of Cheung, et al. [25] showed that a higher average daily time spent using smartphones was significantly associated with both higher physical (bodily pain and general health) and mental (vitality, role limitation due to emotional problems, social functioning, and mental health) health issues.

As such, it appears that the development of unhealthy behaviors, such as excessive ST, may already occur in childhood and persist throughout life. Even though guidelines recommend reducing daily ST, many children overuse ST in real life. To make the point of such cases, Kılıç, et al. [26] found that young children aged 1–60 months were exposed to mobile devices at high levels, and Kabali, et al. [27] found that most children used mobile phones before the age of one. Consequently, a closer look at the factors that influence ST behavior is called for. More specifically, the following factors have been linked to children’s ST: child age, Ethnicity, higher grades, being overweight and obese, parents’ BMI, mother’s weight, parents’ age, parental education, no friends in the neighborhood, loneliness, presence of a TV and a personal computer in the bedroom, parental perception and attitude, high frequency of television and computer use by parents, having limitation rules, unhealthy and healthy food consumption, sleep, physical activity/movement/sports, and eating meals and snacks in front of the TV [28,29,30]. A systematic review showed that the most common correlates of ST in children and adolescents were older age, children’s higher body mass index (BMI), parents’ older age and lower education, higher socio-economic status, lower scores for physical activity, poor quality of the neighborhood, parents’ longer ST, limits, and digital device in the child/adolescent’s bedroom [31]. Another study showed that socio-economic status, eating junk foods, urban residence, aggressive behaviors, obesity, insomnia, a sense of worthlessness, and age had a significant association with longer ST (watching TV and computer use), but with the increasing number of children, the odds ratio of watching TV decreased [32].

While changes in technology have changed families’ schedules all over the world, several more specific reasons were observed as to why Iranian children used DD. First, Iranian society has undergone many changes regarding raising children, for example, parents’ emphasis on having a child who can speak English and use various smartphone applications, and, as a result, bilingual and more gifted children have become more common [33]. In contrast, it appeared that parents are paying less attention to their child’s natural needs, such as parent–child communication, social interaction, or playing outdoors. Unsurprisingly, children scoring high on digital skills and abilities also scored low on competencies with respect to regulating their emotions and communicating appropriately with their peers [34]. Furthermore, for Iranian parents, purchasing a tablet and smartphone for their child is an honor for the parents and shows both a high socio-economic level and (apparent) attention to the child’s needs [35]. Jari, et al. [36] found that 33.4% of Iranian students aged from 6 to 18 years watched TV/video in their leisure time during school days and 53% watched TV/video on holidays. Jari, Qorbani, Motlagh, Heshmat, Ardalan and Kelishadi [36] also found that 47.1% of students in urban areas and 24.2% in rural areas used personal computers during their leisure time. Among the Iranian student population aged 6 to 18 years, Hovsepian, et al. [37] further observed that 18.62% reported more than two hours of ST, 34.11% reported low physical activity, 50.66% reported more than two hours of TV watching, and 9.63% reported more than two hours of computer work. Kabali, Irigoyen, Nunez-Davis, Budacki, Mohanty, Leister and Bonner [27] reported children up to the age of four were able to handle DD on their own and could use them in their bedrooms without parental assistance. Screens may be perceived by parents as a way of improving a child’s learning and making them smarter.

From the overview described above, specific factors related to Iranian culture may explain why (smaller) children might be exposed to longer ST. However, it also appears that the factors mentioned above could not explain the whole range of variance of Iranian children’s ST behavior. Given this, qualitative studies may provide a further and deeper understanding of determinants related to children’s ST. Accordingly, the purpose of the present study was to gain a deeper understanding of the cultural factors in Iranian preschool children’s ST. Accordingly, this study set out to answer the following question: What are the specific factors in the process of excessive ST among Iranian preschool children?

To this end, parents of children aged two to seven were interviewed, and their answers were systematically screened for key themes and subthemes.

## 2. Materials and Methods

### 2.1. Setting and Participants

This qualitative study was conducted between January and March 2020 in Tehran, Iran. The present study aimed to qualitatively explore determinants of excessive ST among Iranian children at kindergarten age, as rated by their mothers. The Tarbiat Modares University Ethics Committee (Approval ID: IR.MODARES.REC.1398.062) reviewed and approved the study, which was performed in accordance with the current and revised version [38] of the Declaration of Helsinki. Participant mothers were fully informed about the aims of the study, the recording of the interviews, and the secure and anonymous data handling. Thereafter, participants signed the written informed consent. Based on the method of purposeful sampling, a total of 20 mothers participated in this study. Semi-structured interviews were carried out with the mothers in separate and secure rooms within kindergartens. The mothers were 30 to 42 years old (*M_age_* = 34.00 years). Fourteen were undergraduated, and seven had a master’s degree. Furthermore, 12 were working part-time or full-time. Their children’s ages ranged from two to seven years (*M_age_* = 3.9 years), and half were girls. Most participants had one child (*n* = 18). Participants were recruited into the study through announcements and events at preschools. After parents were informed of the purpose of the study, the time required, and how to conduct the interviews, a time and date were set for those who volunteered to participate in the study to attend kindergarten and conduct the interview.

### 2.2. Data Collection

An interview date was arranged with interested families by the researcher. The study’s background, aims, and procedures were explained verbally to participants before the interviews. The mothers were also informed that participation was voluntary, and they could opt-out at any time. Finally, each mother signed an informed consent form. In the preschools, the mothers were interviewed separately in separate rooms. We conducted 15 face-to-face interviews, but 5 were conducted over the phone due to the Coronavirus lockdown. To give the mothers a chance to freely describe their child’s and family’s lives, the interview began with an open question (“In your childhood, describe the overall atmosphere of your home and your relationship with your parents”), which was followed by questions to clarify possible and influencing factors. The mothers were asked further questions based on their answers. Among the questions asked were the routines of families and children, typical days of children, interactions between parents and their children, descriptions of children and their homes, etc. The purpose was to gain a deeper understanding of children’s ST behaviors. In addition, the mothers were asked to provide information about their child’s exposure to digital screens and the timing of outdoor/ indoor play: “How much time does your child spend watching television, using a cell phone, and using handheld tablets on an average weekday?”, “How much time does your child spend playing outside and indoors on an average weekday?”, “Describe the day of your child in detail”. Additionally, the mothers were asked if there was anything else they would like to add at the end of each session.

### 2.3. Data Analysis

Using thematic analysis, we analyzed the qualitative data according to the Braun and Clarke approach, which is a flexible, accessible, and theoretically grounded method of analyzing qualitative information [39]. The themes have been investigated at both the semantic and hidden levels. As a result, the following steps were taken: (1) familiarizing with the data, (2) generating initial codes, (3) searching for themes, (4) reviewing themes, (5) defining and naming themes, and (6) producing the report. The interview data were analyzed using MaxQda^®^ version 2018 (VERBI Software, Berlin, Germany). The audiotapes were transcribed verbatim. Coding data in phases was done iteratively in order to create meaningful patterns. Coding and classifications were conducted by the first author (BS) and supervised by two of the other authors (PA and HF). First, we identified codes that represented similar ideas and grouped them together. This allowed us to identify emerging subthemes, which we then combined into single, cohesive themes. We used this process to ensure that the themes we identified were meaningful and could be used to further analyze the data.

## 3. Results

Table 1 shows the 6 themes and 28 subthemes extracted from the interviews. The six themes consisted of (1) temperament/characteristics of the child, (2) parental characteristics, (3) parental health literacy, (4) family psychological atmosphere, (5) home structure, and (6) environmental/social structure of society.

The 28 subthemes consisted of (1) adjusted child, (2) maladjusted child, (3) energy level and mobility rate, (4) child interests and digital capabilities, (5) immature parent, (6) mature parent, (7) narcissistic parent, (8) helicopter parent, (9) parenting pattern, (10) thoughtful parenting, (11) monitoring standards, (12) self-centered parenting, (13) presence of parents, (14) parent–parent interaction, (15) parent –child interaction, (16) the loneliness of the child, (17) different family norms within the extended family, (18) congruency/incongruency of parents with each other, (19) physical space of the home, (20) abundance and accessibility of DD, (21) background TV, (22) child’s ownership of DD, (23) unexpected events and imposed conditions (e.g., COVID-19), (24) climate conditions, (25) environmental requests, (26) facilities and safety of the living environment, (27) kindergarten, and (28) peer group’s role.

Screen-related behaviors were influenced by a number of factors, including family factors (parental and child factors), and environmental and social factors. Based on these findings, we developed a model of factors influencing a child’s screen-related behaviors. In regard to ecological theory, this model provides a comprehensive understanding of relevant elements and their relationships. The model appears at the beginning of the findings section (see Figure 1). Afterward, we discuss the model’s elements and their relationships.

Our findings indicate that the central concept of the model is the family (mesosystem). Considering screen-related behaviors, family life encompasses parental health literacy and family psychological atmosphere (interactions between family members). The child’s and parents’ actions and characteristics can influence family interactions, such as the parent’s characteristics and the child’s temperament/characteristics. A child’s and parent’s behavior is also influenced by social norms, neighborhood and community, workplace, and preschool. Thus, individual family members’ actions are influenced by individual factors at the micro level and environmental/social factors at the macro level.

Children’s and parents’ characteristics influence each other’s behaviors (two-way influence). At the meso level, the sum of interactions determines the structure of families. Family routines, such as watching dinner while watching TV or exchanging screen-related information, are reflected in the digital home structure. An individual’s behavior is affected by the level of their health literacy. Screen use can be influenced and changed by health literacy, which may lead to healthy or unhealthy behaviors. It can affect biological aspects such as energy levels, activity levels, and a child’s interests as well as digital skills. Many families’ attitudes and behaviors contribute over time to the reproduction or changes of socio-cultural norms or alterations to the built environment. Although these effects are complex, other factors such as community policies and economic dynamics, affect them as well. With our study design, we cannot analyze this association, which is why it is depicted as a dashed arrow. Our model describes a process that enables a better understanding of determinants and how they influence a child’s screen behaviors. Using the analyzed interviews, we describe the main categories of the model—the themes—and their relationship.

### 3.1. Screen-Related Themes in Iranian Families

We present the main themes in the bio-ecological model framework. As mentioned, six themes were identified within three domains: Microsystem level (child temperament/characteristics, parent characteristics), Mesosystem level (family psychological atmosphere, home structure, parents health literacy), and Macrosystem level (environmental and social structure of society).

#### 3.1.1. Temperament/Characteristics of Child

In interviews, it was found that both a child’s own ST behaviors and a parent’s ST behaviors are influenced by biological factors. The temperament of a child is a notable biological factor. Parents, especially mothers, are often more stressed when their children are active, feisty, or naughty: [*We let my daughter use a cellphone at least a few hours a day because she’s so annoying and feisty* (mother of a four-year-old)]. In contrast to children with well-adjusted temperaments, maladjusted children may be very active, fussy, angry, and anxious and have intense reactions to a variety of situations. They may also have irregular sleeping and eating habits: [*Occasionally, they may have irregular sleeping and eating patterns: [He does not eat anything at all and only entertains himself with his father’s or my phone, the only way to feed him is to distract him with a phone *(mother of a two-year-old)]. Some children are inactive by nature, get tired easily, and don’t like physical activity. Less active children are more likely to use new technologies. According to one mother: [*He loves cartoons and mobile phones, he watches cartoons, and he loves food and eating.. Sadly, he does not enjoy sports or physical activities* (mother of a five-year-old)]. Children have a variety of interests, which determine the type and amount of activities they participate in. These special interests of the child are nurtured and cared for by the parents. According to one of the mothers: [*He prefers cartoons over other things. He wants to watch cartoons for three or four hours. But the drawing may take half an hour or an hour* (mother of a four-year-old)]. A child’s digital capabilities play a role in the use of DD in addition to their interests. A child’s digital abilities refer to his or her capability to use various DD and applications, such as using different programs, playing online and digital games, and searching the internet for and downloading applications. A mother said: [*He goes on the Internet and downloads without permission. We don’t have Wi-Fi, but he connects to the data and downloads games using the internet, goes to the market, and searches for games* (mother of a five-year-old)].

#### 3.1.2. Parental Characteristics

Characteristics of parents such as helicopter parenting, maturity, immaturity, and narcissism influence their child’s behavior and their parenting style. A helicopter parent protects their child from harm and limits and encloses them in a framework, which, however, appears exaggerated for the child’s individuality and self-pacing degree of responsibility, and thus, beyond the necessity of protection. One of the mothers mentioned this and said: [*I never play exciting games with my son; I feel he would hurt himself if he jumped. At the park, I warn him, telling him not to play with other children, because they will push you and make you fall or something bad happens* (mother of a four-year-old)]. Mature parents take time for their children, play with them, and feel responsible for them. In this regard, one mother said: [*I don’t like my daughter to play alone; I like to join her and play with her because joint activities strengthen our relationship. Even if I have a lot of studies, which I usually do, I make some time for us to do something together* (mother of a four-year-old)]. The child is seen by an immature parent as an opportunity to demonstrate that they are a good parent. It appears that they do not view the child as an independent individual with a unique personality and try to instill their beliefs in them. A mother mentioned: [*After playing with her for half an hour, my husband says I want to play with my phone. It is rare for him to take our daughter to the park or take her out; he usually says I am not in the mood* (mother of a three-year-old). The other characteristics of an immature parent are being rigid, bored, selfish, clingy, appeasing, punishing, indulgent, neglectful, perfectionist, unsupportive, irresponsible, and negligent to the child. Another mother said: [*When my daughter goes to her father, she says “Dad, dad, dad” four or more times, but he doesn’t answer nor pay attention to her* (mother of a five-year-old)]. Parents who are narcissistic do not want children and do not have a positive view of having them. According to these parents, having children can limit their relationships and lead to a loss of opportunities. Their job is more important to these parents than their child. Regarding the loss of their job position, a mother said: [*I didn’t decide to have a child; I don’t like children at all, and I got pregnant at my mother’s insistence.]. My opinion is that having a child is difficult. I get nervous when I hear a child crying* (mother of a five-year-old)].

#### 3.1.3. Parental Health Literacy

Parental health literacy refers to parents’ knowledge and skills of the correct and scientific methods of child-rearing. There are many reasons why DD are used. In terms of using these devices for themselves or their children, parents have different goals. Parents impose fewer restrictions when DD are used for educational purposes: [*The games she plays with her phone are mostly English, animal, alphabet, and fruit name training games*. *She watches cartoons on TV that are educational, such as brushing and personal care, so I think it’s okay and she can watch as much as she wants* (mother of a three-year-old)]. Parents used DD to entertain their children due to a lack of familiarity with age-appropriate games, boredom, different occupations, etc. As these devices are attractive to children, parents can work uninterrupted while their children watch cartoons: [*If I want to start something, I often turn on the TV for my son to watch cartoons while I work* (mother of a two-year-old)]. Parents’ health literacy also refers to their knowledge of the proper use of DD and their disadvantages. When a mother or father controls the use of DD and is aware of its disadvantages, they provide a positive role model for their children. A parent who constantly uses DD in front of their child and plays online games and is very interested in them is not only a good role model but also encourages the child to do so: [*From morning to night, my husband watches movies or plays with his mobile phone, the TV is always on and he plays a lot of online games, he doesn’t have a framework and plays only online games* (mother of a three-year-old)]. In some cases, parents do not have monitoring standards for how much the child uses DD, so the child uses them as much as they like and watches whatever they want. Regarding the lack of content restrictions, a mother said: [*My son enjoys watching war movies and horror films* (mother of a six-year-old)]. In some cases, parents supervise their children when they use these devices, do not leave them alone, and set standards for the children. They teach their children how to use DD correctly; they also use strategies such as removing extra DD from the home and not installing games on their mobile phones. In this regard, one of the mothers said: [*Since I said at the beginning, no game can be installed on my phone, he only take my phone to watch kindergarten channel and see photos* (mother of a three-year-old).

#### 3.1.4. Family Psychological Atmosphere

Children’s behavior is greatly influenced by the psychological atmosphere in the home. Children’s screen time is strongly influenced by the elements in the psychological climate of the home according to the interviews analyzed. The psychological atmosphere of a home includes the presence of parents, family norms within the extended family, the loneliness of the child in the home, parent–parent interaction, parents’ congruency or incongruency of parents with each other, and parent–child interaction.

Parents’ presence is the amount of time they spend at home with their children. Long work hours, having multiple jobs to cover living expenses, and a high volume of work-related tasks often leave parents with little time to spend together and with their children: [*When my husband comes back at seven o’clock, he does not play with my son at all, because he is very tired. There are too many people to deal with every day, and he’s tired of it* (mother of a four-year-old)]. In addition to parental employment, moving to big cities and living in apartments, or home space can also make the child feel lonely: [*For the job, we moved here. We don’t have any family or friends here. The neighbors in the apartment where we live have no young children of the same age as my son, nor do my friends and acquaintances* (mother of a two-year-old)]. In some cases, the extended family can assist parents in raising their children, but in other cases where the family norms conflict with extended family norms, misbehavior in children (such as excessive ST) can result. In the following participant’s response, it is evident that the child is reluctant to return home because she is comfortable watching CDs at her grandmother’s house: [*When I go to work, my daughter stays with my mother and sister. After work, she doesn’t return, but she still likes to be there because they buy her CDs and let her watch; she enjoys being there* (mother of a three-year-old)]. Parental interaction is another important factor in a home. If the parents don’t interact positively and constructively with one another, the child will be affected. Parents’ interactions with each other affect their parenting styles and ultimately the child’s development. Without interaction or intimacy, the child is left alone at home: [*Both my husband and I use phones. We both check Instagram for a while, and my husband also checks his cell phone. We don’t discuss our daily problems or future plans.* (mother of a five-year-old)]. Parent–child interaction is another factor that affects a child’s behavior in addition to parental interaction. This indicates the satisfaction of parents and children with the quality of their interaction with each other. When parents don’t know the child’s age-related needs, they cannot treat the child according to his developmental stage and they are not interested in playing with the child. It is common for these parents to use DD as a form of entertainment for their children: [*Unfortunately, I don’t pay attention to my daughter’s demands, I tell her I’m really busy at the moment and can’t play with you. As soon as she returns from the nursery, I explain to her that I have to make dinner and clean the house, so do not ask me to play with you. When I can’t play with her, I turn the TV on for her to watch* (mother of a five-year-old)]. Another mother pointed out: [*My son and I don’t spend much time together. Since he was three years old, I have only spent money on him and we rarely interact. I surf internet at home and my son is alone in his bedroom because I don’t play with him nor do joint activities with him* (mother of a five-year-old)].

#### 3.1.5. Home Structure

Children’s ST is affected by the home structure, which includes all the elements in the physical space of the home. TV background, the presence and abundance of DD in the home, the physical space of the home, and the child’s ownership of DD all influence screen-related behavior. Television in the background occurs when parents or children are not directly using the television and it is on for other family members (not for the child): [*I Our TV is always on and we watch TV for a long time. After my husband comes home in the afternoon, the TV is still on until bedtime* (mother of a two-year-old)]. Children have access to a variety of DD, so their ability to use these devices is formed at an early age due to the availability of these devices. When parents work from home or have older siblings, the number and variety of DD will increase. According to one of the mothers: [*Besides my mobile phone, I also have a work phone and a laptop at home, and sometimes. I work from home* (mother of a five-year-old)]. Another mother mentioned: [Most of the time, *he plays games on my phone. He enjoys playing some games on his brother’s tablet. The moment the TV is turned on, he wants to switch channels and watch the show. He also wants to play with his brother’s PS4* (mother of a three-year-old)].

According to the interviews, the physical space of a home (e.g., arrangement of furniture, lack of a yard, and limited space in the house) can affect the child’s physical activity. This creates the conditions for the child to use DD. The arrangement of the devices allows for easy access to the TV from anywhere in the house, so the TV is used more often during other activities (e.g., eating). Referring to this, one of the mothers said: [*As usual, when we have dinner, we turn on the TV and watch a series, since the table is in front of the TV* (mother of a two-year-old)]. Another mother emphasized the limited space for the child’s active activities and said: [*Our house is small, we only have one room, so transportation is limited* (mother of a two-year-old)]. It is clear from the interviews that if children have their own DD, they use them more often and it is more difficult for parents to deal with their ST. In this regard, one of the mothers said: [*He was very, very interested in the tablet, we bought it, but we often didn’t get along with it and he watched for hours* (mother of a five-year-old)].

#### 3.1.6. Environmental and Social Structure of Society

This theme relates to the conditions and institutions outside the home that parents have no control over. It includes subthemes such as unexpected events and imposed conditions, climate conditions, environmental requirements, facility and living environment safety, kindergarten, and peer roles. Unexpected events and imposed conditions such as epidemic diseases are a factor that parents avoid in social situations to protect themselves and their children from contamination, thereby limiting social interaction. In addition, epidemics change the working conditions of parents, and many of them are working from home and remotely (online and offline). The situation has changed somewhat due to COVID-19 and the children staying at home: [*You can’t go out with them that often, everyone’s in the house, the parents are worried, and finally you have to leave the kids watching TV* (mother of a two-year-old)]. In addition, parents ensure that their children spend more time at home when the weather is unfavorable to protect them from illness and injury. As a result, they lose the opportunity to interact with their peers. When children stay at home and interactions are reduced, they have an opportunity to use DD. In this regard, one of the mothers mentioned: [*Because of the cold weather, my son goes out less often, maybe once in two to three weeks; he goes shopping with his father. When the weather was nice in the summer, I would take him outside, but now that in the winter everyone is inside and the kids don’t go outside* (mother of a three-year-old)]. Another socio-cultural factor is environmental requests, which point to the demands that today’s digital life places on people. It reflects the norms that have emerged in society regarding children’s digital skills. A mother pointed out that the child would be different from the others and said: [*Children now have access to these digital devices. I can’t take it from my daughter because if I do, she won’t have the skills that her peers have. I know when kids have tablets they stop communicating with each other, they stop talking to each other and the experience they should have isn’t formed, but I can’t tell my daughter not to look, I will never do that. Because my daughter would be different from the others* (mother of a three-year-old)]. As another social/ environmental factor, the living environment includes opportunities that allow parents and children to spend time outside the home. Mentioning the possibility of recreational facilities and the difficulty of accessing these spaces, one of the mothers said: [*recreational spaces are not near us. In our neighborhood there is no park, play area, or place to take the kids. It’s difficult for me to take the children to play because I have to drive a long distance* (mother of a two-year-old)]. One of the other things mothers mentioned was a feeling of insecurity and distrust towards the neighbors: [*It’s not safe for children to play outside or go to neighbors. Safety is a concern around us, and I think it’s become a concern everywhere. I don’t think it’s appropriate for kids to play outside with friends* (mother of a two-year-old)]. Another subtheme was the role of the kindergarten. Today, kindergartens play an important role in the life of children. Since the mothers work, children spend a lot of time in kindergarten. However, television and animation in kindergarten arouse children’s interest in this context. In this regard, one of the mothers said: [*The kindergarten plays cartoons for the children, and my son was very interested in them. Two or three times a day, he watches the animation CDs, and each time takes an hour. Now, unlike before, he watches TV all day* (mother of a five-year-old)]. In addition, children of all ages are strongly influenced by their peers. Peers can influence a child’s behavior and desires and encourage the child to be interested in ST. One mother explained: [*Some of my daughter’s friends have a tablet, and recently my daughter asked for one* (mother of a two-year-old)]. Sometimes, the absence of peers leads the child to spend more time at home and alone, which lays the foundation for the use of DD. One of the mothers pointed out: [*He has no friends except my nephew. When he is alone, he watches cartoons for hours* (mother of a four-year-old)].

## 4. Discussion

The objective of this study was to investigate the determinants of excessive ST in Iranian children aged two to seven years. We interviewed mothers to identify the factors that impacted children’s ST. This study found that the use of digital media by Iranian children was formed and continued under the influence of a number of factors. These factors included the temperament/characteristics of the child, parental characteristics, parental health literacy, family psychological atmosphere, home structure, and the environmental/social structure of society. Children’s screen-related behaviors in daily life are affected by both individual and environmental factors at the micro, meso and macro levels, as revealed by the interviews.

### 4.1. The Micro Level

At the micro level, child and parent characteristics had a significant impact on excessive ST in Iranian children. Other studies have shown a correlation between child characteristics and the use of DD [40,41,42]. Children’s characteristics either lead them to use more DD [43,44] or cause parental fatigue, inability, and despair with respect to managing their behavior, as well as increased parental stress. As a result of dysfunctional interactions between parents and children, parenting styles change. Parenting styles are influenced by three factors: parental personality, child characteristics, and sources of stress and support in life [45,46]. Parental behavior is influenced by child and parent characteristics, according to the Belsky Process Model [47]. Based on this model, parental personality is directly related to parenting, but it is also linked to the child’s characteristics. While parents’ personalities predispose them to certain behaviors, the results depend on a number of factors, including the child’s characteristics. Therefore, parents’ use of DD to calm their children can be a parenting style that allows their children to use DD over a longer period of time. Numerous studies have demonstrated the relationship between parenting styles and the use of these media by children [48,49,50,51]. After analyzing the results of this study, it can be said that the child’s characteristics affect not only how they are perceived by others and their parents, but also how they behave and deal with situations. In addition, children who are naturally sedentary, dislike physical activity, and are easily fatigued are more likely to use DD. This finding is consistent with the results of the research by Arundell, et al. [52] that showed a child’s interests and preferences play a role in increased use of DD. The roles of individual factors have also been described in the LIFES model (Levels of Interacting Family Environmental Subsystems) [53]. The LIFES model posits that biological factors are related to factors constituting parent–child interactions, such as parenting, parental cognitions, and parenting practices, and factors constituting family interactions, for example, family functioning, and family practices such as eating meals in front of the television. Biological factors such as the parents’ and the child’s temperaments also affect factors related to family life at the meso level.

### 4.2. The Meso Level

The main element with the highest number of subthemes were family-related factors (mesosystem). It is not possible to examine the child’s behavior without considering the family and the dominant environment, since the behavior of family members as a whole affects each family member. Child- and parent-related factors at the micro level affect both the media use by the individual and other family members. According to the themes, this effect is created or reinforced by interaction with another theme at the meso level (parental health literacy, family psychological atmosphere, and home structure). Parental character not only plays an important role in parenting style but also in parental health literacy, which influences a child’s development [47], as well as other behaviors, such as using DD.

Analysis of the interviews revealed that the parents’ attitudes towards digital media, their function as role models, and monitoring standards were important subthemes of health literacy. Parents’ attitudes are strongly influenced by the cultural definition of a good parent and good/smart child. Parents usually treat the child in a certain way (authoritarian parenting style) in order to gain society’s approval (good mother/father/good child). This finding is in line with other studies [54,55,56,57]. Regarding the influence of culture on parents’ beliefs, Määttä, et al. [58] showed that parents with a low educational background face greater social pressures related to the time their child uses DD, high costs of sports activities, purchasing and using different DD and learning from them at a young age. Parents with low health literacy not only do not monitor the amount of usage and the watched content, but also do not have specific criteria for buying cartoon CDs and downloading apps. Meanwhile, other studies have shown that having a standard is related to children’s use of DD [59,60,61].

Thomas, et al. [62] have shown that a favorable family environment enables optimal development of a child and vice versa. This theme shows the role and importance of the presence of the father/mother and their interaction with each other and with the child at home. In many cases, long working hours, having multiple jobs to cover living expenses, and heavy workloads result in parents not having enough time to be with their children and spend quality time with them. In addition to the employment of the parents, moving to the big cities and living in an apartment also lead to the child becoming lonelier. While Sisson and Broyles [63] showed in their study that parent–child communication plays a role in the frequency of children’s use of DD, the present study also showed that in addition to the reduction in interaction, the type of parent–child interaction (inappropriate interaction) may also play an important role in increasing or decreasing ST in children. In addition to parent–child interaction, parental interactions and parental conflicts also play an important role in creating an appropriate/inappropriate family atmosphere. Many studies have shown that parental interaction has an impact on children and their behavior [64,65,66,67].

Studies have shown that stress can negatively affect parenting styles and responsiveness [43,68]. Therefore, it can be said that parental conflict can affect the nature of the parent–child relationship and can cause behavioral problems in the child by increasing the stress in the psychological atmosphere of the family. In stressful situations, the presence of the extended family (grandmother/grandfather) can be helpful as a source of support [69]. Our data showed that one of the factors that can affect the psychological atmosphere of the Iranian family is the extended family. Sometimes the presence of the extended family can help the parents in caring for the child, but in some cases, when family norms are at odds within the extended family, this presence, rather than being constructive, may play a role in generating inappropriate behaviors such as the child’s use of DD.

According to the interviews, the housing structure (physical living space, frequency and accessibility of DD, background television, and the child’s ownership of DD) plays an important role in the child’s use of DD. This finding is consistent with the results of other studies that showed that factors related to the home environment affect children’s ST [59,70,71,72].

### 4.3. Socio-Cultural Factors

The environmental/social structure of society refers to the conditions and institutions outside the home space in which parents have no part. It is more related to the social environment and its institutions (e.g., unexpected events and imposed conditions (e.g., COVID-19), climate conditions, environmental requests, facilities and security of the living environment, kindergarten, and the role of peers). The environmental/social structure of society has a significant impact on people’s lifestyles and can both directly and indirectly affect the child’s use of DD. These findings are consistent with other studies that have shown that social interactions restricted by pandemic diseases exacerbated the overuse of DD for socializing [73,74,75]. Therefore, parental interaction, parent–child interaction, family and child daily life, and parenting styles may change under the influence of these difficult and stressful conditions and be a factor in the increased use of DD in children. Weather conditions affected the children’s ST by restricting social interaction between children. This finding has also been shown in other studies to be an influencing factor for the use of these devices by children [4,76,77,78]. Another finding showed that the safety and quality of the neighborhood are influential factors in the children’s use of DD, which is consistent with the findings from other studies [28,29,59,79]. These factors can make it difficult for children to engage in physical activities and play outside the home. Other studies have shown that children who are prevented from playing outside the home by their parents show a greater tendency to use DD [70,79,80]. Another socio-ecological factor is environmental requests. Environmental requests illustrate the demands that life (nowadays) places on people. In fact, changes in society and people’s lifestyles have led to digitalization becoming an inseparable part of people’s lives. Using digital tools to act and live in today’s society has become a skill. If there is insufficient information in this area, a person may depend on others and be considered unqualified. For this reason, parents allow their children to use mobile phones. Parents believe that by not using DD, the child will be different from others and will not acquire the skills necessary for life. In Iranian culture, unlike Western culture, babysitters are not common, and children go with their parents to all places intended for adults (e.g., parties, ceremonies, funerals, and religious ceremonies). In such situations, where the child needs to be quiet and there is no possibility to pursue another activity (physical, cognitive), another kind of environmental demand leads the parents to allow the child to use their mobile phone/tablet for hours. The role of environmental demands or social norms in children’s increasing use of DD can be examined in terms of their influence on parental beliefs and decision-making. Research has shown that making decisions for another person (e.g., parents’ decisions about their children’s health) presents a unique situation in which the perceived social pressures of others are an important factor in making that decision [81,82]. The analysis of the interviews showed that children who go to kindergarten use DD frequently. The children of working parents wait hours beyond the usual and established hours for their parents to return to and from kindergarten, and due to the fact that class time ends, these children spend extra hours in one of the classrooms. They get together and watch children’s programs and animation. This has caused many children in kindergarten to become interested in watching animation. Kindergarten rules, such as that the children can bring their own tablets to kindergarten, lead other children to ask their parents for tablets or mobile phones. Additionally, kindergartens send daily pictures and videos of children and the activities they carry out in the kindergarten to virtual kindergarten networks (such as Instagram, Telegram, and WhatsApp), resulting in children and parents being able to spend more time each day watching them. Parents’ views on the quality of kindergarten are based on the influence of the society in which they live and also on personal experience. Unfortunately, in our culture, parents express their satisfaction with the virtual content that the kindergarten produces (e.g., sending photos), without paying attention to whether their child has acquired these skills or not. This superficiality of parents has encouraged more kindergartens to produce virtual content (just sending photos). Other studies have shown that kindergarten attendance can be effective in lowering the use of DD among children [83,84]. However, our results indicated that the quality of the kindergarten plays a more important role in the child’s use of DD than the care.

Therefore, factors related to environmental/social structure can act as a barrier to children’s physical activities outside the home. In addition, parents’ goals and values are influenced by the socio-cultural context [85]. According to intercultural research, cultural social, and economic factors also undeniably influence beliefs and attitudes [86].

Although the study provided valuable insight into a deeper understanding of the determinants related to Iranian children’s ST, it has the following limitations: While this study aimed at exploring parents’ views, the majority of interviewees were mothers. Furthermore, the samples were recruited only from parents attending kindergartens in a district. Given this, we cannot generalize the pattern of results, which should be interpreted with caution. In addition, the qualitative nature of the data does not allow proof of causality. Improving health education programs and interventions is important to overcome excessive ST in preschool children. These efforts can help parents manage their children’s ST. Future studies may design educational programs for parents. Such educational programs may consist of, but not be limited to, information to increase parental awareness of children’s ST, the advantages and disadvantages of exposing children to ST, including the teaching relevant skills to improve the quality of parent–child and family interactions, and create a healthy lifestyle. Such skills and strategies may need to be tailored to the parenting style and family structure. Future studies should also consider intervention research that demonstrates effective strategies for targeting the determinants and influencing factors of ST in children. Moreover, they may explore cultural differences in the factors influencing children’s use of ST and examine both mothers’ and fathers’ points of view with diverse socio-economic conditions.

## 5. Conclusions

This study found that Iranian children’s use of digital media emerges and persists under the influence of a broad range of factors. These factors included the child’s temperament/characteristics, parental characteristics, parental health literacy, family psychological atmosphere, home structure, and the environmental/social structure of society. It is not possible to examine the child’s behavior without considering the family, since the behavior of family members as a whole influences each family member. The main factor, which has also provided more resources, is family-related factors. The interviews revealed that both child-related and parent-related factors influence the extent of media use by the children and other family members. Depending on the themes gained, this effect is created or reinforced through interaction with other themes such as the psychological atmosphere of the family and the physical structure of the home. In general, the characteristics of parents and children as a microsystem can affect the psychological atmosphere of the home and the structure of the home as a mesosystem, ultimately leading to the emergence or reinforcement of behaviors such as excessive ST in the child. The child’s excessive ST, which arises under the influence of the micro- and mesosystem factors, in turn, also affects them (two-way interaction). Parents’ behaviors and attitudes, and family communications and interactions contribute to healthy ST habits in children. Additionally, children’s increased use of new technologies can be influenced by the environmental/social structure (unexpected events and imposed conditions (e.g., COVID-19), climate (weather) conditions, environmental requirements, facilities and safety of the living environment, kindergarten, and peer group’s role). Therefore, many factors such as individual and child-related factors and factors related to family, society, and culture are involved in ST as a multidimensional behavior, and more than one factor is instrumental in its emergence and continuity. Thus, in order to change or manage a child’s behavior (ST), it must be viewed in the context of the family, and lifestyle changes must be implemented. In summary, interventions should make parents aware of their role in reducing children’s ST, and interventions can benefit from considering parental perceptions of children’s behaviors.

## Figures and Tables

**Figure 1 children-10-01193-f001:**
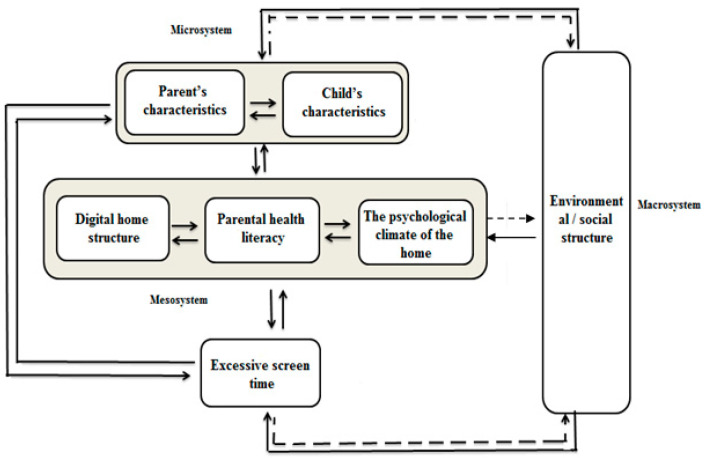
Influential factors and child’s screen time.

**Table 1 children-10-01193-t001:** Codes, subthemes, and themes extracted from interviews.

Codes	Subtheme	Theme
Empathetic child	Adjusted child	Temperament/characteristics of the child
Good communication		
Ability to play alone		
Interaction ability		
Angry child	Maladjusted child	
Anxious child		
Mischievous/naughty child		
Not interested in playing		
Irregular sleeping and eating habits		
Level of fatigue	The energy level and mobility rate	
Level of mobility		
Online playing	Child interests and digital capabilities	
Using the Internet		
Playing digital games		
Using different apps		
Interest in different media		
Eating meals and snacks with TV		
Limiting mother	Helicopter parent	Parental characteristics
Controlling parent		
Self-sacrificing mother		
Over-responding to child’s wishes		
Self-respect	Mature parent	
Patient father		
Responsible mother		
Playing with the child		
Well-informed mother		
Joint activity with the child		
Strict parent	Immature parent	
Bored mother		
Selfish mother		
Clingy mother		
Appeasing mother		
Punishing mother		
Indulgent mother		
Neglecting the child		
Perfectionist mother		
Non-supportive mother		
Irresponsible and negligent father		
Not respecting the personality of the child		
Not interested in children	Narcissistic parent	
Priority of Job to child		
Priority of relationships to child		
Not a priority	Parenting pattern	Parental health literacy
Overuse		
Managed usage		
Explain to the child		
Excessive use by parents		
Don’t use it in front of the child		
Install the games for the child		
Playing digital games with the child		
Working with DD at home		
Not playing with the child	Self-centered parenting	
Entertaining the child with DD		
Learning and teaching	Thoughtful parenting	
Physical and psychological damage		
Restrictions and supervision	Monitoring standards	
Lack of criteria for buying CDs (CD: Compact Disc)		
No limit to the time and content		
Absent parent	Presence of parents	Family psychological atmosphere
Busy parent		
Not responding parent		
Not cooperating with parents	Different family norms within extended family	
No restrictions in extended family		
Migration	The loneliness of the child	
No playmate/ friend		
Positive interaction	Parent–parent interaction	
Reduced interaction		
Non-intimate relationships		
Cold relationship		
Reduced interaction	Parent–child interaction	
Appropriate interaction		
Inappropriate interaction		
Personality conflicts	Congruency/Incongruency of parents with each other	
Cultural differences		
Harmony in child rearing		
Contradiction in education		
Limited space	Physical space of the home	Home structure
A yard at home		
Arrangement of home equipment		
Variety of DD	Abundance and accessibility of DD	
Too many DD		
Availability of DD		
Using it in front of the child		
Use by others	Background TV	
TV being on while playing		
TV being on without purpose		
Waiting for favorite TV program		
Tablet ownership	Child’s ownership of DD	
Mobile phone ownership		
TV in the children’s bedroom		
Ownership of SIM card with internet		
Doing work at home during COVID-19	Unexpected events and imposed conditions	Environmental/social structure of society
Decreased social communication during COVID-19		
Staying at home	Climate condition	
Less communication with peers		
Society norms	Environmental requests	
Special situations		
Cultural facilities	Facilities and security of the living environment	
Mistrust of neighbors		
Feeling insecure in living area		
Distance to recreational facilities		
Watching cartoons	Kindergarten	
kindergarten rules		
Digital content production		
No peers	Peer group’s role	
Digital games with peers		
Peer influence on children’s requests		

## Data Availability

Data are made available to explicit experts in the field. Such experts should clearly formulate their hypotheses; further, they should fully describe, how and where the do securely store the data file, and how they make sure that the data file is not shared with and securely protected from third parties.

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
