# Peer review of "Why Do Iranian Preschool-Aged Children Spend too Much Time in Front of Screens? A Preliminary Qualitative Study"

_children, 2023, doi:10.3390/children10071193_

Round 1

Reviewer 1 Report

First of all I would like to congratulate the authors for the work. However, the article needs to be improved. 

The Abstract needs to be summarised. It is too long. 

From the introduction (line 106, 110) delete 2 and 3. Also add the objective, research hypothesis and research questions at the end of the introduction. 

In the procedure section, the process followed has not been described. How the analysis population was contacted, etc. 

Add the limitations and future perspectives derived from this research.

Lines 625,626,627,628 and 629 delete them. 

All references are wrong. They should be brought into line with the journal's rules. It is also necessary to add a current bibliography. Here are some researches that can help you: 

Sanz-Martín, D.; Ubago-Jiménez, J.L.; Ruiz-Tendero, G.; Zurita-Ortega, F.; Melguizo-Ibáñez, E.; Puertas-Molero, P. The Relationships between Physical Activity, Screen Time and Sleep Time According to the Adolescents' Sex and the Day of the Week. Healthcare 2022, 10, 1955. https://doi.org/10.3390/healthcare10101955 

Liebherr, M.; Kohler, M.; Brailovskaia, J.; Brand, M.; Antons, S. Screen Time and Attention Subdomains in Children Aged 6 to 10 Years. Children 2022, 9, 1393. https://doi.org/10.3390/children9091393 

Panjeti-Madan, V.N.; Ranganathan, P. Impact of Screen Time on Children's Development: Cognitive, Language, Physical, and Social and Emotional Domains. Multimodal Technol. Interact. 2023, 7, 52. https://doi.org/10.3390/mti7050052 

Cheung, M.-c.; Lai, J.S.K.; Yip, J. Influences of Smartphone and Computer Use on Health-Related Quality of Life of Early Adolescents. Int. J. Environ. Res. Public Health 2022, 19, 2100. https://doi.org/10.3390/ijerph19042100

English level is Ok. Minor errors have been detected

Author Response

Dear Reviewer,

Thank you very much for all your kind efforts.

We have addressed all concerns raised by the Reviewers. Please see the detailed point-by-point-response attached as a separate file. 

Again, thank you very much for the care devoted to our manuscript. 

Reviewer 2 Report

Very interesting, useful and relevant work, but the scientific depth must be improved.

I propose the following topics, that in my opinion will improve the manuscript.

1.     Where is the answer to the title question?

In the conclusion (and abstract) must be noted that the parents bear the heavy and important responsibility of reducing SТ.

2.     Where are the quantitative data claimed in the title?

Improvement: The authors can change the title. For example:

Why Do Iranian Preschool-Aged Children Spend too much Time in Front of Screens? A Preliminary Study

3.     The abstract does not contain a sufficent description of the results and conclusions.

The Abstract and Conclusions must be improved.

Improvement. Based on the proposed corrections and others, these two paragraphs should be rewritten clearly and in depth. In Abstract must be included the main conclusions and the authors must underline the own contributions. If the authors change the title – they can focus the discussion to the factors influencing on the Iranian Preschool-Aged Children to Spend too much Time in Front of Screens. The unwritten rule is that most readers only look at these paragraphs – abstract and conclusions.

4. The hypothesis is, in my opinion, very general and not well defined.

Improvement. It can be rewritten clearly and concrete.

I hope that the proposed corrections will increase the quality of the manuscript and possibly its citability.

Author Response

(The authors gave the same response as above.)

Round 2

Reviewer 1 Report

Dear authors, 

The changes have been made successfully. 

Reviewer 2 Report

In future - the authors can increase the participants and must include the fathers. From 20 mothers the authors can not obtaine clear conlussion for targetted question.

In my opinion the children answers to the targetted question are very interesting too.